# The Multiple Roles of Sphingomyelin in Parkinson’s Disease

**DOI:** 10.3390/biom11091311

**Published:** 2021-09-05

**Authors:** Paola Signorelli, Carmela Conte, Elisabetta Albi

**Affiliations:** 1Biochemistry and Molecular Biology Laboratory, Health Sciences Department, University of Milan, 20142 Milan, Italy; paola.signorelli@unimi.it; 2Department of Pharmaceutical Sciences, University of Perugia, 06126 Perugia, Italy; carmela.conte@unipg.it

**Keywords:** sphingomyelin, sphingolipids, Parkinson’s disease, neurodegeneration

## Abstract

Advances over the past decade have improved our understanding of the role of sphingolipid in the onset and progression of Parkinson’s disease. Much attention has been paid to ceramide derived molecules, especially glucocerebroside, and little on sphingomyelin, a critical molecule for brain physiopathology. Sphingomyelin has been proposed to be involved in PD due to its presence in the myelin sheath and for its role in nerve impulse transmission, in presynaptic plasticity, and in neurotransmitter receptor localization. The analysis of sphingomyelin-metabolizing enzymes, the development of specific inhibitors, and advanced mass spectrometry have all provided insight into the signaling mechanisms of sphingomyelin and its implications in Parkinson’s disease. This review describes in vitro and in vivo studies with often conflicting results. We focus on the synthesis and degradation enzymes of sphingomyelin, highlighting the genetic risks and the molecular alterations associated with Parkinson’s disease.

## 1. Introduction

Parkinson’s disease (PD) is the second most common neurodegenerative disease (ND) after Alzheimer’s disease (AD). Recent breakthroughs in our knowledge of the molecular mechanisms underlying PD involve unfolded protein responses and protein aggregation. Of note, dopaminergic neuronal cells of the substantia nigra suffer from an abnormal accumulation and aggregation of a specific presynaptic neuronal protein, the α-synuclein (α-Syn). [1]. As a consequence, neurons undergo degeneration. The degenerative mechanism in reality is complex. Mitochondrial dysfunction, oxidative stress, neuroinflammation, and lysosomal dysfunction have been called into question [2]. Recently, a lot of attention has been given to lysosomes. They are fascinating organelles known in the past for being the recycling centers of the cell and recently considered essential to mediate metabolic adaptation, to control the functionality of organelles, and to regulate nutrient-dependent signal transduction [3]. Due to all these functions, lysosomes maintain neuronal homeostasis in the brain. Thus, their dysfunction underlies NDs, especially PD. In fact, several genes and/or genetic risk factors linked to PD encode for lysosomal proteins. Thus, PD-associated gene mutation is associated with lysosomal impairment. Because α-Syn degradation occurs in lysosomes, their dysfunction affects α-Syn turnover causing an unavoidable increase in its intracellular levels, leading to its accumulation and aggregation in tissues [4]. Studies spanning two decades have revealed the association between PD and lysosomal sphingolipid (Sph) metabolism defects [5]. Rare genetic diseases due the intralysosomal accumulation of Sphs due to the defect of their degradation are called lysosomal storage diseases (LSD). They include gangliosidoses, Krabbe disease, Fabry disease, Farber disease, Gaucher disease, Niemann-Pick disease (characterized by accumulation of gangliosides) (GM), galactosylceramide (GalCer), trihexosyl ceramide (TrihexosylCer), ceramide (Cer), glucosylceramide (GCer) and sphingomyelin (SM). During the last ten years, epidemiologic and biochemical evidence has brought to our attention a link between Gaucher disease, caused by a deficiency of glucocerebrosidase (GBA), which is useful for degrading GCer, and PD. GBA variants predispose individuals to αSyn accumulation and neurodegeneration even in the heterozygous status [6]. Of note, GCer accumulation promotes the conversion of αSyn into a proteinase K-resistant conformation [7]. Recently, it has been demonstrated that inhibition of Cer synthesis reduces αSyn accumulation [8]. 

While there is a plethora of information about the relation between Cer and PD, there is also a dearth of knowledge about the involvement of SM in PD. However, in recent years, the field that studies the roles of SM in the brain has made important strides in understanding the different ways SM can regulate brain physiopathology. 

In this review, we will explore the different behavior of SM and SM metabolism enzymes in PD from gene risk to lipid and enzyme modifications. 

## 2. Sphingomyelin Metabolism

SM is synthesized by SM-synthase using both phosphatidylcholine (PC) as donor of a phosphorylcholine group and free Cer. It is catabolized to Cer by sphingomyelinase (SMase). Cer in turn can be metabolized via different pathways: (1) it is degraded by ceramidase (Cerase), producing sphingosine (SP) that can be phosphorylated by sphingosine kinase (SphK) to form sphingosine-1-phosphate (S1P); (2) it is complexed with sugar molecules to form GCer and GalCer by GCer-synthase (GCerS) and GalCer-synthase (GalCerS), respectively; (3) it is used to form more complex molecules, the ganglioside (GM). GCer and GalCer can be degraded by GBA or galactocerebrosidase (GalBA), freeing Cer and glucose (G) or galactose (Gal), respectively (Figure 1)

### 2.1. Sphingomyelinases

There are several SMase isoenzymes grouped into families in relation to the pH optimum for their activity: acid SMase (aSMase), neutral SMase (nSMase) and alkaline SMase (alkSMase) (Table 1). While alk-SMase, encoded by the ectonucleotide pyrophosphatase/phosphodiesterase 7 (*ENPP7)* gene, is mainly localized in the gastrointestinal tract and to some extent in the liver, aSMase and nSMase are ubiquitous and are located in lysosomes and in cellular membranes, respectively [5]. 

The family of aSMase includes two forms of Zn^++^dependent enzymes encoded by the sphingomyelin phosphodiesterase 1 (*SMPD1*) gene: (1) aSMase associated with the endosomal/lysosomal compartment; (2) secretory aSMase found in the plasma and in the culture medium of cells stimulated by various factors [9]. The endosomal/lysosomal compartment is required during apoptotic processes. Under stress conditions, aSMase rapidly moves towards the outer leaflet of the plasma membrane [10].

The family of nSMases has four isoforms located in plasma and intracellular membranes, including nuclear membranes [11]. They are encoded by different genes: *SMPD2* for nSMase1, *SMPD3* for nSMase2, *SMPD4* for nSMase3, and *SMPD5* for mitochondrial-associated nSMase (MA-nSMase) [5]. nSMase 1 and 2 are both Mg^2+^-dependent and differ in term of their localization and functions. nSMase 1 is located in the in the endoplasmic reticulum and the Golgi apparatus, while nSMase 2 is located in the plasma membrane, in multilamellar bodies, and the in the nuclear membrane. nSMase 1 is activated in response to stress and degeneration and nSMase 2 in response to cell growth arrest, exosome formation, and inflammatory response. nSMase 3 is mostly found in the endoplasmic reticulum of skeletal muscle and the heart but not in the liver and it is involved in inflammation and tumorigenesis. MA-nSMase is the least studied and it is highly expressed in the testis, pancreas, and fat tissue [5].

### 2.2. Sphingomyelin Synthase

Sphingomyelin-synthase (SM-synthase) is a PC/Cer cholinephosphotransferase, an enzyme able to transfer the phosphocholine group from PC onto the primary hydroxyl group of Cer. In this way, it is possible: (1) to generate SM; (2) to free diacylglycerol (DAG) from PC; and (3) to reduce the level of free Cer [12]. Thus, SM synthase regulates the DAG/Cer balance, which is essential for cell fate. A reverse SM-synthase enzyme uses SM as a donor of phosphorylcholine to synthesize PC [13]. The members of the SM-synthase family are controlled by different genes. PC: Cer cholinephosphotransferase 1 (*SGMS1)* encodes for SM-synthase 1 and *SGMS2* for SM-synthase 2. Despite belonging to the same family, the sterile alpha motif domain containing 8 (*SAMD8)* encodes for SM-synthase related protein (SMSr) that does not synthesize SM but SM analog Cer phosphoethanolamine in the endoplasmic reticulum by moving a phosphoethanolamine head group from phosphoethanolamine to Cer and consequently regulating the Cer level [14]. Really, all members of the SM-synthase family are able to synthesize Cer phosphoethanolamine [15].

Both SM-synthase 1 and SM-synthase 2 are expressed in the brain. Of note, SM-synthase 2 localizes in the microdomains that anchor signal transduction proteins of hippocampal neurons [16].

## 3. Sphingomyelin in the Brain

The role of Sphs as bioactive signaling molecules able to regulate cell fate decision puts them at center stage for brain physiopathology. Changes in Sph metabolism have been proposed in neurologic diseases including multiple sclerosis, Alzheimer’s disease, and PD [5]. Among Sphs, SM has both structural and functional roles (Figure 2). This lipid supports brain myelination, a process closely associated with neuronal cell function; forms ordered domains where nicotinic acetylcholine receptor [17] is located; and acts at a nuclear level by regulating chromatin function [18]. Thus, SM is essential for brain development and cognitive abilities [19] and it is altered in neurodegeneration [20]. 

### 3.1. Sphingomyelin in Myelin Sheath

The myelin sheath is a highly specialized multilayer structured membrane that surrounds axons in the central and peripheral nervous system [21]. 80% of its dry weight is composed of lipids, including glycerophospholipids, cholesterol, and Sphs [22]. The last two lipids are rigid molecules and make myelin a structure with low conductance necessary to increase the propagation speed of the nerve impulse along fibers [23]. It has been described for a long time that myelin is produced by Schwann cells in the peripheral nervous system and by oligodendrocytes (OLs) in the central nervous system. In children, human breast milk represents the best dietary choice since it is rich in SM content that facilitates the cognitive development modulated by OLs activity with increased axon myelination [24] (Figure 2). In adults, newly appearing OLs polarize from OL precursor cells (OPCs), using essential lipids in the diet by permitting both neuron plasticity characterized by the ability to make new connections to activate new pathways and to unmask secondary roads, and the remyelination process, which is the regeneration of the myelin sheaths following demyelination [25]. A blend containing docosahexaenoic acid, arachidonic acid, vitamin B12, vitamin B9, iron, and SM increases the number of OPCs and promotes their differentiation and maturation into OLs [26]. More recently, astrocytes have been considered to make a major contribution to the lipids in myelin by playing a role in myelination and remyelination via diffusion of chemical messengers [27]. 

Membranous myelin sheath is particularly rich in hydroxySMs and long-chain SMs content [28]. These lipids are asymmetrically distributed to yield the biophysical properties required for membrane curvature and myelin compaction. A lipid derangement was called into question in the altered myelin structure in neurodegeneration [29]. In fact, hydroxySMs are essential for the long-term stability of the myelin sheath, and their shortage can lead to late onset of demyelination and neurodegeneration [30]. Interestingly, myelin basic protein is not incorporated into the cholesterol-rich domains if SM is absent, indicating that SM is essential for myelin basic protein function [31]. In humans, the low intake of essential lipids for SM synthesis with diet increases demyelination and slows down the remyelination during pathological processes [32]. Fish oil mixture stimulates the human OPC maturation with consequent improving of myelin basic protein, myelin glycoprotein, and SM synthesis [32]. Kim et al. [33] reported that a member of the ATP-binding cassette subfamily A (ABCA) type 8 (ABCA8) is strongly expressed in OL-enriched white matter regions where it significantly stimulates both SM-synthase 1 expression and SM production. Moreover, the authors demonstrated a correlation between ABCA8 expression in the prefrontal cortex and age-associated myelination characterized by upregulation of myelinating gene p25α by indicating the role of ABCA8 in the myelin formation and maintenance via regulation of SM metabolism in OLs [33]. In addition, OLs express nSMase2 and respond to stress by producing Cer with a consequent change in myelin structure and therefore negative curvatures in membranes. nSMase2 genetic deletion or pharmacological inhibition in OLs normalize the Cer content in remyelinated fibers by increasing thickness and compaction [34]. Additionally, the fatty acid (FA) composition of Sphs is extremely important for myelin structure. Reduction of very-long-chain Cer induces neurological abnormalities due to hypomyelination in the brain [35]. 

### 3.2. Sphingomyelin and Derived Metabolites in Brain Cell Function

In brain cells, lysophospholipids, endocannabinoids, and Sphs play different roles as biosignal molecules by participating in the processes of neurogenesis, neuronal plasticity, neuronal selection, impulse conduction [36], and the regulation of numerous ion pumps, channels, and transporters [37]. Of note, SM and Cer play important roles in neurogenesis; their depletion limits the outgrowth of axons [38]. The action of SM mainly depends on their ability to interact with cholesterol to form lipid rafts, specific regions of membrane with a different regulatory role in cellular functions such as neuronal differentiation and synaptic transmission, and neuronal–glial connections [38]. The SM/Cer balance is indispensable for brain physiology [39]. nSMase controls the clustering of N-methyl-d-aspartate receptors that are fundamental for brain development and function due to their ability to regulate the postsynaptic excitatory currents [39]. Moreover, the neuronal postsynaptic function can be modulated by nSMase 2 by changing the levels of SM and Cer [40]. The release of transmitters to the terminals of excitatory synapses is stimulated by SP [41] and the localization of synapsin-I in pre-synaptic compartments is regulated by S1P [41]. Both SP and S1P stimulate the release of glutamate from the synaptosomes [41,42]. 

The culture of HT22 murine hippocampal neuronal cells with decanoic acid (C10), which is able to reproduce medium chain triglyceride in diets consisting of medium chain fatty acids such as ketogenic diets, induces an increase in SM content [43].

### 3.3. Sphingomyelin Metabolism in Brain Pathology

Dysregulation of SM metabolism is an important effector of brain damage [17]. It is involved in inflammatory processes that underlie several diseases of the central nervous system [44]. Changes in SM metabolism have proven to be important for inducing a lipid raft rearrangement with consequent development of neurological diseases [38]. Interestingly, nSMase deficient mice exhibited compromised plasticity [45]. Through this pathway, cells can control Cer level and modify their physiological response, suggesting that manipulation of this novel pathway may have therapeutic implications. 

It is known that the aSMase/Cer balance is altered in chronic stress and major depressive disorder characterized by mood changes, somatic alterations, and often suicide. Therefore, aSMase was studied as a target in the preparation of several commonly used antidepressants [46]. This is relevant considering that aSMase interacts with transient receptor potential canonical 6 ion channels located in the plasma membrane of neurons that act as receptors for antidepressive herbal remedies, i.e., hyperforin [46]. Antidepressant drugs such as amitriptyline and fluoxetine stimulate SM accumulation in lysosomes and Golgi membranes, which are responsible for autophagy of hippocampal neurons, by inhibiting aSMas. This is relevant considering that autophagy is the basis of the neurogenesis promotion with integration of novel neurons into neuronal networks impaired by stress and depression [47]. 

Thus, SM accumulation is considered a positive factor of neurogenesis. On the other hand, Hsiao et al. demonstrated a high level of SM-synthase 1 expression in the hippocampus of *post mortem* brains in AD patients with an increase in amyloid-beta (Aβ)peptide generation [48]. Accordingly, inhibition of SM-synthase 1 activity significantly reduced the level of Aβ. Interestingly, enzyme downregulation by siRNA reduced the extracellular Aβ because exosomes were oversecreted, stimulating Aβ uptake by microglia [49]. 

## 4. Sphingomyelin and Parkinson’s Disease 

### 4.1. Genetic Risks

There is accumulating evidence supporting the association between PD and Niemann-Pick disease (NPD), an autosomal recessive disorder responsible for aSMase deficiency. The direct consequence is a lysosomal storage disease characterized by accumulation of SM and its precursor lipids in lysosomes, mainly of macrophages located in different organs responsible for hepatosplenomegaly, cytopenias, lung disease, and neurologic symptoms [50]. In relation to the clinical picture and diagnostic parameters, there are four subtypes of NPD: type A, B, C, and E [51]. Type A is characterized by a dramatic reduction in aSMase volume. It is an infantile neurovisceral form with developmental delay, hepatosplenomegaly progressive neurodegeneration, and death before the age of three. Type B is a less aggressive form with a partial reduction in aSMase. The patients have hepatosplenomegaly, thrombocytopenia, interstitial lung disease, very limited neurological involvement, but usually live into adulthood. Type C results from a mutation in *NPC1* and *NPC2* genes encoding for proteins involved in cholesterol efflux from lysosomes. It is included in this family because of the metabolic interaction between SM and cholesterol. In fact, part of cell cholesterol is linked to SM by van der Waals forces and activation of nSMase is able to dissociate the two molecules [17]. Clinically, it is a heterogeneous form with systemic, neurologic, and psychiatric involvement that usually affects adults. Type E is a variant that affects adults [51]. 

It has been demonstrated that nSMase is upregulated in the hippocampus of aSMase KO mice reproducing NPD ([50]. Pharmacological treatment with glucocorticoid dexamethasone in aSMase knockout (KO) mice stimulates nSMase reducing SM levels in synapses and improving memory and learning capabilities [52].

In the neurons of aSMase KO mice, there is a pathological downregulation of the endocannabinoid system and inhibitors of the endocannabinoid-degrading enzyme fatty acid amide hydrolase reduce SM storage by limiting neurodegeneration [53].

In a study of 287 subjects evaluated as AD patients in the Ashkenazi Jewish population, it has been highlighted that nine patients had NPD SMPD1 mutations. The type of mutation was not significant. Among 400 control subjects, only two presented the SMPD1 mutations. 

The authors tend to consider NPD a risk factor for PD [54]. In the same year, Gan-Or et al. reported that the different SMPD1 mutations had differential effects on the risk for PD [55]. Of note, Mao et al. identified an association between Leu-Ala (Val) repeat variants in SMPD1 and patients with sporadic PD [56]. Moreover, among 198 Chinese PD cases, four rare variants in SMPD1 (p.P332R, p.Y500H, p.P533L, and p.R591C), absent in control subjects, were identified. In the Ashkenazi Jewish cohort of 1592 PD patients, 1.4% carried the p.L302P or p.fsP330 mutation compared with 0.37% in control patients [57]. Interestingly, reduced aSMase activity was associated with the early onset of PD. In the same study, the authors demonstrated an accumulation of α-Syn in SMPD1 KO HeLa and BE(2)-M17 dopaminergic cells. 

Therefore, the studies all reported an association between PD and NPD of type A and B. Ouled Amar Bencheikh et al. wanted to analyze the association between variants of type C NPD that induces defects in cholesterol traffic and PD [58]. The results identified a total of nine common and 126 rare type C NPD variants among 2657 PD patients, but the data analyses did not support a link between heterozygous variants and PD.

### 4.2. Pathogenetic Mechanisms

The exact mechanism by which SM is involved in PD remains controversial. It is evident that SM is differently involved in PD in relation to the localization of its metabolism. Additionally, results of studies conflict with one another over whether they were conducted in in vivo or in vitro systems.

In vivo studies showed that in the anterior cingulate cortex and not in the occipital cortex of PD patients there was a decreased level of SM C23:0, C24:1, C26:1 and an increased level of SM C18:1 and C20:0 [59]. Xicoy et al. reported the depletion of saturated SM species (14:0SM and 16:0SM) in the putamen of post-mortem human brain of PD patients [60]. Interestingly, the iron chelation as useful therapy in the treatment of PD for its neuronal protection action increased 16:0 (SM) [61]. Thus, saturated medium chain SM species appear to be essential for brain protection against PD, probably since the saturated SM species might be more specific for myelin sheath structure than unsaturated SM species. Interestingly, no differences were observed in the content of SM in lipid rafts located in frontal cortex cells of PD patients compared to control subjects, indicating this brain structure remains unaltered in neurodegenerative disease [62].

The possibility that the changes in SM species was due to the alteration of SM metabolism was considered. A down-regulation of nSMase2 in the substantia nigra of PD patients and in the mouse striatum during aging was reported [63]. To understand the mechanism of SM pathway action in dopaminergic cell death, Ruberg et al. [64] studied the behavior of SM metabolism in apoptosis. The authors demonstrated a positive role of free radical signal and the nuclear translocation of NF-kappa B in the apoptogenic SM-dependent transduction pathway [65]. Starting from this study and considering the convergent data from the postmortem brains of PD patients, it is suggested that the relation of the free radical/SM pathway might be the basis of the death of dopaminergic neurons.

On the other hand, *SMPD3* gene expression in PD patients is significantly lower than in healthy subjects [66]. In support of this, nSMase2 deficiency influenced the clinical picture of the early onset of PD [67]. Similar results were observed in a mouse model of PD obtained with 1-methyl-4-phenyl-1,2,3,6-tetrahydropyridine in which a strong downregulation of nSMase has been described [68]. 

Different results were obtained with in vitro studies. In catecholaminergic neuroblastoma cell line SH-SY5Y treated with the catecholaminergic neurotoxin 6-hydroxydopamine to mimic PD, the 16:0SM, 18:0SM, and 24:1SM level was increased [60].

Moreover, SM treatment of neuronal cells induced an overexpression of both ATP-binding cassette subfamily A 5 (ABCA5) transporters, a group of proteins that transport lipids around the brain, and α-Syn, supporting the role of SM in inducing PD [69].

However, the involvement of SM in the pathogenetic mechanism of PD is still debated. There is evidence that cell membrane lipid microdomains rich in SM regulate α-Syn aggregation [70]. In fact, α-Syn interacts with vesicle membranes containing cholesterol and SM [71]. Thus, α-Syn links these lipids to form high-density lipoprotein-like (HDL-like) particles in which α-Syn in part forms a core structure with a broken helical conformation (1–100 residues) and in part is in a random-coil state (40 C-terminal residues) [72]. The transfer of oligomeric aggregates of α-Syn between neuron-like cells was mediated by the release of extracellular vesicles, recently proposed to be involved in cell–cell communication, regulated by SM/Cer ratio [73]. Reduced level of nSMase caused inhibition of Cer production in extracellular vesicles with consequent decreases in the transfer of α-Syn [73]. In support of this, Tsutsumi et al. used GW4869, an inhibitor of nSMase2 to demonstrate a reduction in exosomal release from microglia cells with a limitation in the inflammation process [74]. The authors showed a prevention of dopaminergic neurodegeneration characteristic of PD by IFN-γ/LPS without affecting NO production. Differently, the use of D609 as an inhibitor of SM-synthase and NO synthase did not induce the same effect, despite the strong inhibition of NO synthase, indicating the specific role of nSMase 2 in microglia activation in PD. 

In addition, Park et al. reported that blood–brain barrier (BBB) integrity was critically regulated by aSMase. The enzyme was strongly overexpressed in brain endothelium and/or plasma of both aged humans and mice. It accelerated caveolae-mediated transcytosis with BBB impairment, inducing neurodegenerative change [75].

### 4.3. Sphingomyelin as Marker

Metabolic disorder of SM species is associated with autophagy and synucleinopathy and their increased concentration in the blood of patients was considered a marker of neurodegeneration [76].

In PD patients, an increase in SM level in the blood serum was reported [77]. The author believed that it was due to the degradation of myelin without degradation of SM for inhibition of its enzymatic pathway. Xicoy et al. found a positive concordance between PD and blood levels of saturated very long chain SM species present in lipoproteins (SM 26:0) [60]. The lipidomic analysis of cerebrospinal fluid (CSF) of PD patients showed a high level of SM species, among other lipids. The neuropathological classification of patient’s stage included Lewy body disease limbic stage (LBDL), Lewy body disease early neocortical stage (LBDE), and Lewy body disease neocortical stage (LBDN). Of note, SM species were most significantly increased in LBDL [78].

## 5. Conclusions and Perspectives

The complexity and the speed of the metabolic mechanisms makes SM extremely difficult to study, particularly in the context of the brain. Here, the SM is essential for myelin sheath, impulse transmission, synaptic plasticity, location of neurotransmitter receptor, and BBB integrity, as reported above. Much has been accomplished in terms of defining the changes of the multifaceted aspects of SM in PD. However, the results reported in the literature are not completely clear due to the diversity of the experimental models and the methods used. These results indicate that specific perturbations regard aSMase and nSMase. A schematic representation of our knowledge of the role of SM metabolism disorder in PD is shown in Figure 3.

From the analysis of these studies, it appears that a reduction in aSMase, due to either genetic disease (NPD) or gene downexpression/protein downregulation, correlates with PD. 

In PD, the reduction in aSMase leads to excessive accumulation of SM, which becomes “toxic” to the cells. On the other hand, the reduction of nSMase is associated with a reduced release of exosomes containing α-Syn. Since once it is in the extracellular space, α-Syn is removed by proteolysis by extracellular enzymes, a failure to release results in accumulation. Furthermore, the reduction of nSMase is responsible for anomalous neuronal plasticity and reduced neurogenesis. 

Thus, even today the role of SM and its metabolism in PD is not fully understood and a crucial point remains the defect of neurogenesis that might be useful to limit the damage. 

It has been demonstrated that in mouse embryonic stem cells, the deletion of sirtuina-1 deacetilasi NAD-dipendente results in an accumulation of SM due to the sphingomyelin phosphodiesterase acid like 3B (SMPDL3B) reduction, impairing neural differentiation in vitro and in vivo [79]. Interestingly, embryonic hippocampal cell differentiation is stimulated by vitamin D3 [80], which links its receptor to nuclear lipid microdomains rich in cholesterol and SM [81]. After linking with its receptor, vitamin D3 stimulates cell differentiation by activating nSMase. 

Thus, a possible mechanism to limit PD damage might be to stimulate stem cell differentiation via the SM pathway.

## Figures and Tables

**Figure 1 biomolecules-11-01311-f001:**
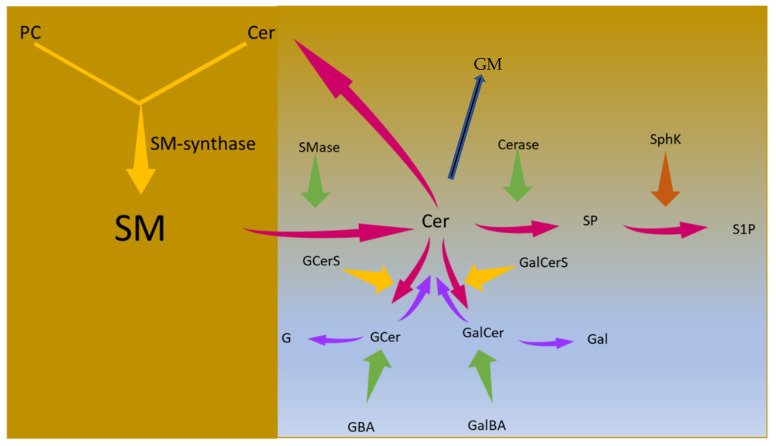
Sphingomyelin metabolism in the brain. Description in the text. SM, sphingomyelin; PC, phosphatidylcholine; Cer, ceramide; SMase, sphingomyelinase; Cerase, ceramidase; SphK, sphingosine kinase; SP, sphingosine, S1P, sphingosine-1-phosphate; GCerS, glucosylceramide synthase; GalCerS, galactosylceramide synthase; GCer, glucosylceramide; GalCer, galactosylceramide; GBA, glucocerebrosidase; GalBA galactocerebrosidase; G, glucose; Gal, galactose.; GM, ganglioside. The yellow arrows indicate the metabolic syntheses; the amaranth red arrows represent the production of Cer from SM and its utilization; the green arrows indicate enzymes with degradation function and the orange arrow an enzyme with phosphorylation function; the purple arrows indicate the effect of GBA and GalBA, i.e., production of Cer and G or Gal; the blue arrow indicates the use of Cer to synthesise GM.

**Figure 2 biomolecules-11-01311-f002:**
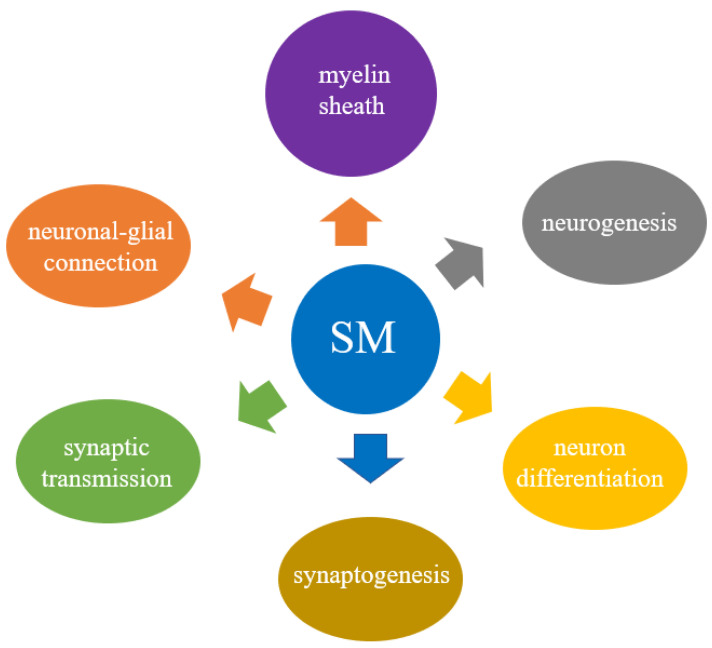
Schematic representation of the main roles of sphingomyelin in brain physiology. Description in the text (Section 3.1 and Section 3.2).

**Figure 3 biomolecules-11-01311-f003:**
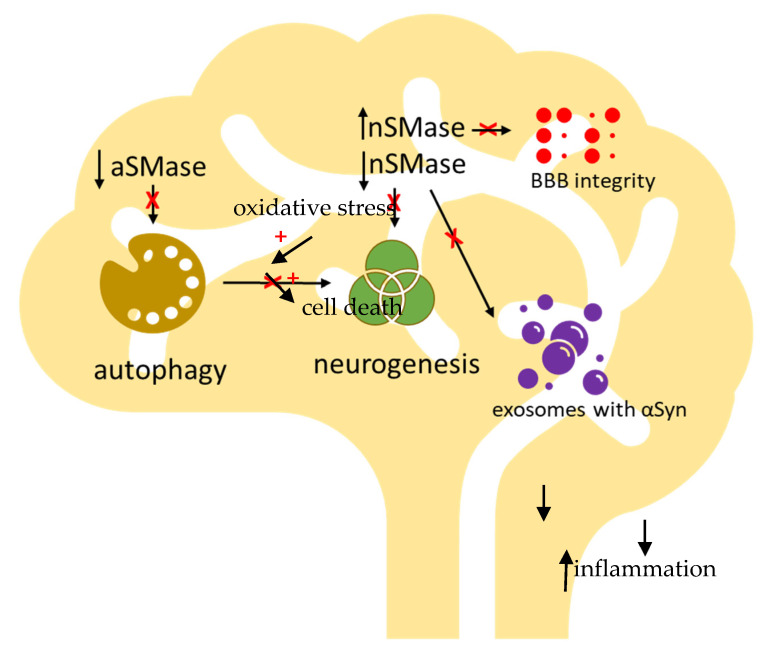
Schematic representation of main alterations induced by the SM metabolism disorder in PD. Reduction of acid sphingomyelinase (aSMase) impairs autophagy, which is a process necessary to allow neurogenesis to occur. Activation of aSMase by free radicals is responsible for cell death. Reduction of neutral sphingomyelinase (nSMase) is responsible for the inhibition of stem cell differentiation, hindering the neurogenesis, and for the slowed release of exosomes containing αSyn, limiting its degradation by extracellular enzymes and stimulating inflammation.

**Table 1 biomolecules-11-01311-t001:** Genes encoding for sphingomyelinases. *ENPP7*, ectonucleotide pyrophosphatase/phosphodiesterase; SMPD, sphingomyelin phosphodiesterase; alkSMase, alkaline sphingomyelinase; aSMase (s), secretory sphingomyelinase; aSMase (e/l), sphingomyelinase associated with the endosomal/lysosomal compartment; nSMase, neutral sphingomyelinase; MA-nSMase, mitochondrial-associated neutral sphingomyelinase.

GENE	PROTEIN	TISSUE	LOCALIZATION
ENPP7	alkSMase	gastrointestinal	cell membranes
SMPD1	aSMase	ubiquitous	(s) plasma and culture medium (e/l) endosome/lysosome
SMPD2	nSMase1	ubiquitous	endoplasmic reticulum Golgi apparatus
SMPD3	nSMase2	ubiquitous	plasma membrane multilamellar bodies nuclear membrane
SMPD4	nSMase3	skeletal muscle heart	endoplasmic reticulum
SMPD5	MA-nSMase	testis pancreas fat tissue	mithocondrian

## Data Availability

Not applicable.

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
