# Peer review of "The Multiple Roles of Sphingomyelin in Parkinson’s Disease"

_biomolecules, 2021, doi:10.3390/biom11091311_

Round 1

Reviewer 1 Report

In this review the authors have summarized the findings concerning the role of sphingomyelin and the enzymes involved in its metabolism, in Parkinson disease. The topic is interesting and still little debated, giving to the authors the opportunity to provide a comprehensive  overview and their critical position on this matter.

However, the authors should bring some changes to the structure and contents of the manuscript to make it more informative and – scientifically - user-friendly.

  • The authors should examine more in depth the potential role of SM in the mechanisms involved in Parkinson disease pathogenesis, such as inflammation, mitochondrial dysfunctions…etc., so suggesting how alterations in SM expression/content and its metabolism could contribute to PD development beyond the current evidence. The Figure 4 should be updated accordingly.
  • The figures are in general few explanatory. In particular, in the Figure 1 the authors should explain the meaning of the color of the arrows they used. Moreover, to be consistent with the text, the authors should clearly indicate in the figure the pathways that they mentioned in the manuscript (the third one is missing). The authors should replace the Figure 2 with a table reporting , genes encoding SMase enzymes, the corresponding proteins and their tissue and cellular localization. Of note, this figure is wrongly cited in the text. Similarly, Figure 3 does not provide any value added and could be replaced by a figure summarizing the SM role in the brain physiology, in general.
  • The section “ Conclusion and perspective” should be rewritten since it is redundant. The reference to antidepressant drugs (line 353) and psychic stress (line357) should be clarified.

Minor issues:

  • The authors should clarify the sentence “Gaucher patients are ….status” at lines 48-50, since Gaucher disease is caused by homozygous mutations in the GBA gene.
  • The author should add a reference for the sentence “Changes in Sph metabolism…,PD.” (lines 129-131.
  • The sentences at lines 249-251 seems inappropriate in relation to the context. The authors should eliminate them or explain why they have included them in this part of the manuscript.
  • The authors should carefully reread the text and correct potential typing and oversight errors, and punctuation.
  • The references 60 and 76 are the same.

Author Response

In this review the authors have summarized the findings concerning the role of sphingomyelin and the enzymes involved in its metabolism, in Parkinson disease. The topic is interesting and still little debated, giving to the authors the opportunity to provide a comprehensive  overview and their critical position on this matter.

However, the authors should bring some changes to the structure and contents of the manuscript to make it more informative and – scientifically - user-friendly.

  • The authors should examine more in depth the potential role of SM in the mechanisms involved in Parkinson disease pathogenesis, such as inflammation, mitochondrial dysfunctions…etc., so suggesting how alterations in SM expression/content and its metabolism could contribute to PD development beyond the current evidence. The Figure 4 should be updated accordingly.

Thank you very much for this observation. Accordingly, the text has been changed (lines 314-320, 349) and also figure 4

  • The figures are in general few explanatory. In particular, in the Figure 1 the authors should explain the meaning of the color of the arrows they used.

It has been made (lines 79-82)

Moreover, to be consistent with the text, the authors should clearly indicate in the figure the pathways that they mentioned in the manuscript (the third one is missing).

In fig. 1 the GM was included

The authors should replace the Figure 2 with a table reporting , genes encoding SMase enzymes, the corresponding proteins and their tissue and cellular localization. Of note, this figure is wrongly cited in the text.

Figure 2 has been replaced with a table and citation in the text has been included (line 86)

Similarly, Figure 3 does not provide any value added and could be replaced by a figure summarizing the SM role in the brain physiology, in general.

Figure 3 has been deleted and has been replaced with the required figure (figure 2 in the present version)

  • The section “ Conclusion and perspective” should be rewritten since it is redundant. The reference to antidepressant drugs (line 353) and psychic stress (line357) should be clarified.

The section “Conclusion and perspectives” has been revised

Minor issues:

  • The authors should clarify the sentence “Gaucher patients are ….status” at lines 48-50, since Gaucher disease is caused by homozygous mutations in the GBA gene.

The statement has been revised (line 49)

  • The author should add a reference for the sentence “Changes in Sph metabolism…,PD.” (lines 129-131.

The reference has been added (line 148)

  • The sentences at lines 249-251 seems inappropriate in relation to the context. The authors should eliminate them or explain why they have included them in this part of the manuscript.

It has been revised (lines 186, 187 in the present version)

  • The authors should carefully reread the text and correct potential typing and oversight errors, and punctuation.

It has been made

  • The references 60 and 76 are the same.

It has been corrected

Reviewer 2 Report

In this review manuscript by Signorelli et al., the authors try to summarize the current understandings of roles of sphingomyelin (SM) in the pathology of Parkinson disease (PD). The contents are of importance, because this manuscript focuses on SM. Although the roles of SM in PD are not fully understood and the literatures are controversial, the authors provided a nice summary and the manuscript is easy to follow. I have relatively minor concerns as follows.

  1. On page 1, lines 37–39, alpha-syn is degraded via the autophagic-lysosomal pathway and the UPS. Is the phrase “mostly in lysosomes” correct?
  2. On page 2, lines 60–68, please include a brief explanation of these pathways in the figure legend, and/or can the authors indicate each pathway in the figure? This will help the readers following the SM metabolism.
  3. On page 5, lines 171–172, what is the correlation between the stress response and “consequent negative curvature”?
  4. On page 6, lines 238–239, please explain about the metabolic interaction SM-cholesterol.
  5. On page 7, lines 278–279, PD is a neurodegenerative disorder which is characterized by neuronal loss, why “probably for their role in myelin sheath”?
  6. On page 8, lines 305–307, alpha-syn may be secreted in association with exosomes from neurons, how microglial exosomes relate to transcellular spreading of alpha-syn? Regarding tau, microglial exosomes may be involved in the propagation of pathological tau (Asai, NatNeurosci, 2015, doi:10.1038/nn.4132).
  7. On page 8, lines 312–315, the nSMase-BBB section is interesting, but does a loss of BBB integrity occur in PD? Please add some background of BBB dysfunction in PD, or, neurodegenerative diseases, if possible.
  8. On page 8, the section 4.3, use of SM as a PD biomarker is interesting. The authors mentioned SM in the blood or CSF samples. SM is insoluble and present in the microdomains or in the myelin sheath, and is also contained in lipoproteins such as HDL. The altered SM levels in PD patients reflect the altered composition of lipoproteins?
  9. The authors mentioned that alpha-syn is degraded via the autophagic/lysosomal pathway, but not further mentioned in the 4.2 “Pathogenic mechanisms” section. Is this OK?
  10. On page 10, lines 378–379, regarding “stimulate stem cell differentiation”, the neuronal loss in PD occurs in the striatum, but adult neurogenesis occurs in the hippocampus. Thus, it is not clear how stimulation of stem cell differentiation relates to the protection of the SM pathway against PD pathology. Please explain.
  11. On page 9, lines 345–377, it seems that there are too many line breaks.
  12. Please fix typos, such as “disfunction” on page1, “Zn++” on page 3, etc. On page 3, line 86, “The endosomal/lysosomal” will be “The endosomal/lysosomal compartment”?

Author Response

In this review manuscript by Signorelli et al., the authors try to summarize the current understandings of roles of sphingomyelin (SM) in the pathology of Parkinson disease (PD). The contents are of importance, because this manuscript focuses on SM. Although the roles of SM in PD are not fully understood and the literatures are controversial, the authors provided a nice summary and the manuscript is easy to follow. I have relatively minor concerns as follows.

  1. On page 1, lines 37–39, alpha-syn is degraded via the autophagic-lysosomal pathway and the UPS. Is the phrase “mostly in lysosomes” correct?

Monstly has been removed

  1. On page 2, lines 60–68, please include a brief explanation of these pathways in the figure legend, and/or can the authors indicate each pathway in the figure? This will help the readers following the SM metabolism.

The figure legend has been revised

  1. On page 5, lines 171–172, what is the correlation between the stress response and “consequent negative curvature”?

It has been clarified (lines 198, 199)

  1. On page 6, lines 238–239, please explain about the metabolic interaction SM-cholesterol.

It has been explained (lines 266-269)

  1. On page 7, lines 278–279, PD is a neurodegenerative disorder which is characterized by neuronal loss, why “probably for their role in myelin sheath”?

The statement has been revised (lines 308-310)

  1. On page 8, lines 305–307, alpha-syn may be secreted in association with exosomes from neurons, how microglial exosomes relate to transcellular spreading of alpha-syn? Regarding tau, microglial exosomes may be involved in the propagation of pathological tau (Asai, NatNeurosci, 2015, doi:10.1038/nn.4132).

I’m sorry, I did not read papers on the role of SM in exosomes released from microglia and related to Tau protein in PD

  1. On page 8, lines 312–315, the nSMase-BBB section is interesting, but does a loss of BBB integrity occur in PD? Please add some background of BBB dysfunction in PD, or, neurodegenerative diseases, if possible.

Yes, the disfunction of BBB in PD and the role of aSMase is reported in the paper n° 75 (lines 355-358)

  1. On page 8, the section 4.3, use of SM as a PD biomarker is interesting. The authors mentioned SM in the blood or CSF samples. SM is insoluble and present in the microdomains or in the myelin sheath, and is also contained in lipoproteins such as HDL. The altered SM levels in PD patients reflect the altered composition of lipoproteins?

Yes, it has been reported (line 366)

  1. The authors mentioned that alpha-syn is degraded via the autophagic/lysosomal pathway, but not further mentioned in the 4.2 “Pathogenic mechanisms” section. Is this OK?

Yes, no specific paper on the relation SM-autophagic/lysosomal pathway-alpha-Syn in PD is present

  1. On page 10, lines 378–379, regarding “stimulate stem cell differentiation”, the neuronal loss in PD occurs in the striatum, but adult neurogenesis occurs in the hippocampus. Thus, it is not clear how stimulation of stem cell differentiation relates to the protection of the SM pathway against PD pathology. Please explain.

In PD there is also alteration of hippocampus with memory loss [references: 68-80]

  1. On page 9, lines 345–377, it seems that there are too many line breaks.

It has been revised

  1. Please fix typos, such as “disfunction” on page1, “Zn++” on page 3, etc. On page 3, line 86, “The endosomal/lysosomal” will be “The endosomal/lysosomal compartment”?

Thank you very much, they have been revised